# Light intensity-induced photocurrent switching effect

Agnieszka Podborska[1], Maciej Suchecki [1,2], Krzysztof Mech [1], Mateusz Marzec [1], Kacper Pilarczyk [1✉] & Konrad Szaciłowski [1]

A better control over processes responsible for the photocurrent generation in semi-conductors and nanocomposites is essential in the fabrication of photovoltaic devices, efficient photocatalysts and optoelectronic elements. Therefore, new approaches towards photochemical properties tuning are intensively searched for. Among numerous parameters, the photocurrent polarity is of great importance to the overall performance of a device. Usually, the polarity is controlled through an alignment of electronic states/bands, tailoring of applied potential or suitable selection of incident light wavelengths. In most scenarios though, the influence of light intensity is somehow neglected and either some arbitrarily chosen, natural conditions are mimicked or this parameter is varied only in a narrow range. Here we present a ternary nanocomposite in which the persistent photocurrent polarity switching is achieved through changes in the light intensity. We also present arguments suggesting this behaviour is of a general character and should be considered also in other photochemical systems.

---

[1] Academic Centre for Materials and Nanotechnology, AGH University of Science and Technology, al. A. Mickiewicza 30, 30-059 Kraków, Poland. [2] Faculty of Physics and Applied Computer Science, AGH University of Science and Technology, al. A. Mickiewicza 30, 30-059 Kraków, Poland. ✉email: kpilarcz@agh.edu.pl

The studies on new promising materials for applications in photovoltaics, photocatalysis and optoelectronics are flourishing and numerous groups focus on semiconductor-based systems interacting with light in their works. One of the key aspects of these research efforts is an attempt to pinpoint mechanisms responsible for the photocurrent generation and to tune photoelectrochemical properties appropriately[1]. Numerous articles have been published on this topic and various phenomena have been already discussed—particularly, processes influencing the intensity and polarity of photocurrents have been presented on numerous occasions[2–4]. We also have contributed to this field and explained how the electrical response to the irradiation may be tuned in nanocomposites and how the output of a simple optoelectronic device made of hybrid materials could be treated in terms of information theory[5–7].

We have discussed, among others, the utility of the photoelectrochemical photocurrents switching (PEPS) effect in the design of optoelectronic logic gates and chemosensors[8]. This phenomenon—which basically consists in changing the photocurrents polarity through variations in the applied potential or the incident light wavelength—have been observed for a variety of neat semiconductors (e.g. titanium (IV) oxide, cadmium sulphide, bismuth orthovanadate, etc.)[9–11], but may be also induced by an appropriately designed surface modification with organic compounds (e.g. anthraquinone derivatives)[12] and/or carbon nanostructures[13].

Intriguingly, in the latter case—i.e. with the use of two modifiers—some emergent features tend to appear either due to interactions between molecular species or collaborative/competitive interplay with semiconductors surface. Generally speaking, changes introduced to the electronic structure of such ternary hybrid material may affect not only the energy levels alignment but also the kinetics of charge carriers transfer, the rate of electron/hole trapping/detrapping events and other physicochemical properties which cannot be easily tuned with the use of only one modifier. This approach, although not typically applied, offers a new insight into materials modification.

At the same time, there are several reports on a light intensity-dependent anomalous decrease in the photocatalytic activity of metal-doped wide bandgap semiconductors, e.g. $Pt@TiO_2$[11], $ZnO$[14], $GaN:ZnO$ solid solutions[15], or plasmonic heterostructures[16]. In the first case, the effect was attributed to "interactions between the Pt surface and reactive intermediates (…) at low light intensity condition"[11]. Interestingly, electron trapping processes at the metal co-catalyst were proposed as an origin of this phenomenon[17]—an idea closely related to the reasoning discussed later in the text. Moreover, a similar effect of the photocurrent superlinear dependence on incident light intensity was associated by Klee et al. with the competing influence of various recombination centres[18] and a decreased rate of charge carriers recombination at high light fluxes[19]—a similar concept is presented in this article.

In the presented studies we focus on the photochemistry of a thoroughly examined wide bandgap semiconductor—zinc oxide (ZnO)—and possible ways to influence its response to the irradiation. Based on our previous experience and information found in the literature[20,21] we use chloranilic acid (3,6-dichloro-2,5-dihydroxy-1,4-benzoquinone, here denoted as CA) as a promising electron acceptor in the attempt to affect the electron propagation through the material. We also try to introduce various carbon modifiers to enable trapping/detrapping processes and to control—to some extent—the kinetics of charge carriers transport in the system[22]. Surprisingly, we obtain the most interesting results for the ternary hybrid material composed of ZnO, CA and fullerenols—i.e. hydroxylated fullerenes. In such nanocomposite we observe not only the ordinary PEPS effect controlled through

shifts in the applied potential but also an intriguing phenomenon—light intensity-induced photocurrent switching (LIIPS), i.e. the change of the photocurrent polarity caused by the adjustment of incident light intensity—which may be of great importance in the photochemical and photophysical studies of both neat semiconductors and various nanocomposites.

## Results

**The first experiments**. While testing various combinations of quinone derivatives and carbon nanostructures used as modifiers for ZnO with particular interest in the processes responsible for the photocurrent generation, we have recorded multiple photocurrent spectra with varying electrode potentials and incident light wavelengths in a standard procedure (cf. "Methods" section). At that time, we noticed a peculiar behaviour for electrodes covered with a ternary hybrid material containing ZnO, CA and fullerenols ($C_{60}(OH)_{30-36}$, here denoted as COH), which have been previously subjected to the irradiation (with 365 nm diode) at the arbitrarily chosen potential of $-0.3$ V vs. Ag/AgCl reference electrode. We have observed that the photocurrent polarity depends on the incident light flux—low intensities result in the cathodic output and high intensities yield the anodic photocurrent (Fig. 1). The detailed analysis of this unusual response (Supplementary Fig. 1) allows one to pinpoint the exact light intensity value, at which the transition occurs.

The ITO@PET electrode covered with CA–COH@ZnO ternary hybrid material was irradiated with a decreasing light ($\lambda = 365$ nm) intensity at $-300$ mV vs. Ag/AgCl in oxygen-rich electrolyte.

In the course of experimental work we have defined three conditions, under which this effect occurs—both modifiers (i.e. COH and CA) are present in the system, the hybrid material has to be subjected to prolonged irradiation (light soaking) prior to the photoelectrochemical measurement and the electrolyte has to be equilibrated with air during the experiment (i.e. it contains dissolved oxygen). The first requirement indicates some sort of interaction exists between molecular species, which pushed us towards a detailed investigation on this matter (vide infra). The second one suggests that some kind of light-driven transformation occurs and this step influences significantly charge carrier transport within the nanocomposite—at the same time, the electrode potential applied within the conditioning step may be varied in a relatively broad range (vide Fig. 2). Noteworthy, for samples conditioned without irradiation (i.e. with the potential applied for 10 min in the dark) no LIIPS effect has been observed (Supplementary Fig. 2). The third one reveals the important role oxygen species play in the generation of cathodic photocurrents during the experiment. Based on the first observation we have focused on the mixtures of COH and CA seeking for any changes in spectroscopic data.

**The interplay between modifiers**. First of all, we have noticed some significant deviations in FTIR spectra (Supplementary Fig. 3) with the most pronounced alteration in bands at ~1600 cm$^{-1}$ which may be associated with C=O bond stretching mode originating from CA. In the presence of COH a shift towards higher wavenumbers may be noticed for one of the peaks, which in turn could be interpreted as a new interaction involving one of the carbonyl groups—the most probable scenario here is the formation of hydrogen bond engaging hydroxyl moiety from COH.

In order to verify this assumption, we used DFT calculations testing different configurations (including various available tautomers of CA) of both molecular species linked with a hydrogen bond (H-bond) or a covalent bond (C-bond) through an oxygen bridge. We obtained good results for a model based on a CA structure presented in one of the reviews on

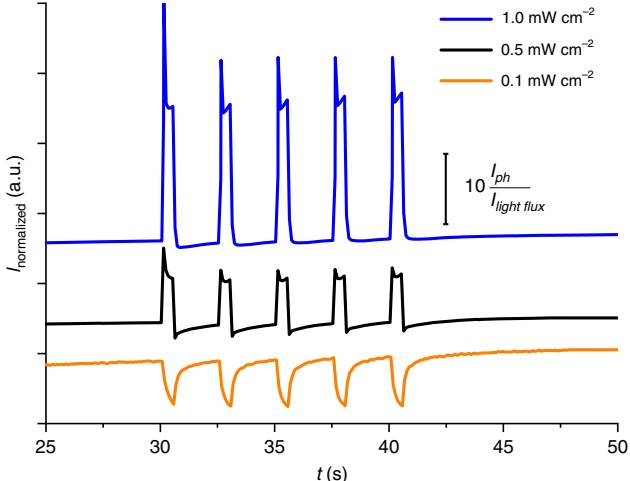

**Fig. 1** The result of a preliminary experiment in which light intensity-induced photocurrent switching (LIIPS) is demonstrated.

1,4-dihydroxybenzoquinone derivatives[23] connected through oxygen atoms from carbonyl groups (CA side) with hydrogen atoms from COH hydroxyl moieties (H-bond; Supplementary Fig. 4a) or through an oxygen atom with a carbon atom from COH (C-bond; Supplementary Fig. 4b). The simulation of UV–Vis spectra for the proposed systems revealed an additional absorption band at ~550 nm in the former case (H-bond), which presence is in agreement with the calculated HOMO–LUMO transition of a charge transfer character and was confirmed experimentally for a freshly prepared sample (Supplementary Fig. 4c). The most interesting part is, the discussed transition vanishes when the modifiers become connected through the covalent bond (C-bond), not only in the simulation, but also in the UV–Vis spectra recorded for the mixture subjected to irradiation for more than 10 min (Supplementary Fig. 4d). This is a strong indication that the chemistry of the system changes due to the interaction with light.

Further analysis of UV–Vis spectra recorded for individual modifiers and their mixture for varying pH (Supplementary Fig. 5) exposed an increased tendency towards agglomeration of COH in the presence of CA—conclusion drawn from the appearance of the absorption band at ~265 nm at pH = 0[24]. The origin of this transition has been additionally verified through a comparison between UV–Vis spectra recorded for fresh mixtures of CA and COH and the aged ones (solutions left for 2 weeks at room temperature in the dark)—in the latter case the absorption at 265 nm was more pronounced (Supplementary Fig. 6a). Interestingly, when the solution is irradiated in a similar manner to the hybrid material conditioning process, we may observe a comparable change in the UV–Vis spectra profile (Supplementary Fig. 6b). Furthermore, the disappearance of the absorption band at ~550 nm can also be noticed in both scenarios (i.e. aging and irradiation), which suggests (according to the analysis of simulated UV–Vis spectra) formation of a covalent bond between CA and COH. This effect turns out to be important for the mechanism responsible for the LIIPS effect occurrence, as it is discussed further in the text.

**The hybrid materials**. Now, if we consider neat ZnO, an n-type semiconductor of ~3.2–3.3 eV wide direct bandgap[25,26], we observed a typical behaviour reported in numerous articles and purely anodic response to the irradiation with photon energies exceeding the optical bandgap width. Significantly, no signs of the photoelectrochemical photocurrent switching effect

(Supplementary Fig. 7a, b) have been noticed. The only distinctive feature of the ZnO sample revealed in this study, compared with the data found in the literature[27,28], is its relatively low flat-band potential value which was determined equal to −0.92 V vs. NHE (Supplementary Fig. 8) and a rather low energy of the Fermi level with respect to the valence band edge—the energy gap $\Phi_{BP}$ determined based on the UPS measurements was equal to 1.8 eV (Supplementary Fig. 9a). These discrepancies may result however from the surface restructuring caused by the homogenization of material prior to the measurements (cf. "Methods" section).

The interesting changes can be observed upon modification of ZnO either with CA or the system composed of both CA and COH. Noteworthy, in the case COH is used alone, the separation of materials occurs indicating that without mediating influence of CA the COH interaction with ZnO surface is negligible—a result which correlates well with a lack of any significant differences in the electronic structure of ZnO and COH@ZnO determined based on the UPS measurements (the only change is the increase in the interfacial dipole layer barrier by ~0.2 eV in the latter system)—hence we do not discuss samples COH@ZnO any further. At the same time, CA coordinates to the surface—most probably through adjacent hydroxyl and carbonyl groups[12,29]—and strongly affects not only charge carriers (electrons) concentration (as derived from the slopes shown in Supplementary Fig. 8) but also induces the photocurrent polarity change for oxygen-rich electrolyte and negative electrode potentials (Supplementary Fig. 7c, d). It is generally consistent with a strongly acceptor character of CA and the mechanism behind the photocurrent switching may be of a similar nature as discussed in our previous works[12,13]. Additional, clear evidence for this new, strong interaction between ZnO surface and modifiers may be found in FTIR spectra—significant shifts and/or partial quenching of absorption bands originating from hydroxyl and carbonyl groups of CA may be noticed (Supplementary Fig. 10).

A puzzling thing here is, although we clearly see additional absorption bands in the UV–Vis spectra of composites (Supplementary Fig. 11), we have not observed any photosensitisation in the photocurrent action spectra. For the system containing both modifiers we recorded amplified cathodic photocurrents and a slight, anodic shift (ca. 0.1 V) of the switching potential (i.e. the potential value at which the photocurrent polarity is reversed, Supplementary Fig. 7e, f). At the same time, we saw some major discrepancies between three investigated materials when the response to irradiation was measured in the Kelvin probe configuration (i.e. the determination of surface photovoltage, Supplementary Fig. 12)—although in all the cases we observed profiles typical for an n-type semiconductor, their shapes differ (especially in the case of CA–COH@ZnO system) which, together with slight variations in relaxation times, may be associated with the change in charge carriers accumulation processes in the presence of COH. Last but not least, when two modifiers are used at the same time we recorded sharp signals also for longer wavelengths, an observation which seems to be consistent with the proposed HOMO–LUMO transition in CA–COH part of the composite which also leads to charge redistribution.

**The LIIPS effect**. The aforementioned change in charge carriers accumulation and transport mechanisms caused by the presence of additional electron trapping states leads to a peculiar response to the irradiation in the case of composite materials. We have already mentioned that the most pronounced changes may be noticed in the oxygen-rich electrolytes—particularly, the strong amplification of cathodic photocurrents occurs when CA is present in the system and this effect is additionally enhanced upon

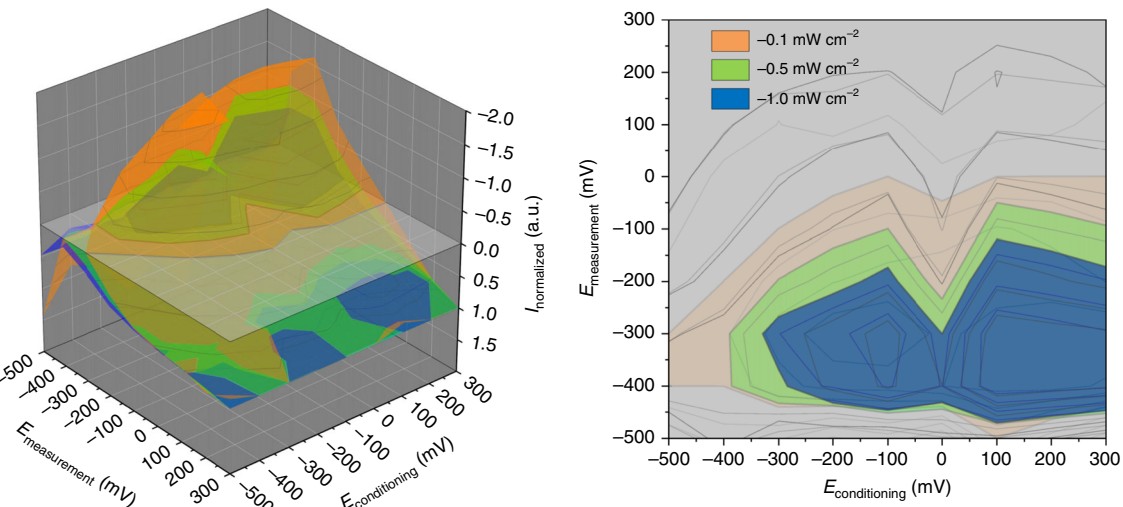

**Fig. 2 The map illustrating how different experimental conditions affect the photocurrents polarity for the CA–COH@ZnO hybrid.** The electrode potential during the pre-treatment step and the measurement itself, as well as the light intensity were varied. Different colours (right) indicate regions of cathodic photocurrents at various incident light intensities. It may be easily noticed, the lower photon flux is, the more cathodic photocurrents "island" expands.

the addition of COH. This observation is in agreement with previous investigations of TiO$_2$, CdS and PbMoO$_4$ modified with CA or various anthraquinone dyes[12,13,30,31]. In these systems, depending on the applied potential, oxidation of water or reduction of dissolved oxygen can occur, which leads to the potential/wavelength photocurrent switching.

When we analysed the raw data in detail, we noticed that in the short-wavelength region an additional photocurrent switching takes place for CA–COH@ZnO sample which cannot be explained based solely on the energetics of the system. The only reasonable cause could come from the light source characteristics, which is not flat—in terms of the intensity—in this region. For that reason, we decided to investigate the response of neat ZnO and its hybrids to the irradiation at 365 nm with the applied potential equal to −300 mV (which corresponds to the conditions at which we observed the switching for the first time) for three light intensities. It turned out that the LIIPS effect is exclusively observed in the case of the ternary hybrid (Supplementary Fig. 13) and it requires the sample to be irradiated while being subjected to the potential from a certain potential window prior to the main measurement (Fig. 2).

Noteworthy, the LIIPS magnitude may be controlled in several ways—e.g. through the irradiation time during the conditioning step (when the time is shorter than 10 min the photocurrent polarity switching typically is less pronounced for the intermediate light intensities; thus we called this light flux range, where the system behaviour is hard to predict and switching may be incomplete, transitional). Moreover, in the course of further research we were able to prove the phenomenon is of a persistent nature, as its manifestation does not depend on the light intensity changes direction (Supplementary Fig. 14) and it cannot be fully reversed even with the extensive cyclic voltammetry treatment within an accessible electrochemical window (−0.4/+0.5 V vs. Ag/AgCl; although it is affected to some extent in the transitional intensity range; Supplementary Fig. 15).

We may state at this point, the effect is not caused merely by some sort of charge accumulation, but rather is a consequence of the hybrid material transformation specific to the ternary system. Based on the already presented data we came to the conclusion, that a prolonged interaction with light leads to the modification within the CA–COH pair[32] anchored to the ZnO surface—the

most likely scenario involves the formation of a new covalent bond between both modifiers, leading to the creation of an additional pathway for charge carriers transport which contributes to the generation of cathodic photocurrents. The footprint of this transformation can be noticed as a new peak in the differential pulse voltammogram (DPV) at approximately −400 mV vs. Ag/AgCl (Supplementary Fig. 16).

To verify whether any structural changes occur when modifiers are present in the system or after the conditioning procedure is applied to the ternary material, scanning electron microscopy was used (Supplementary Fig. 17). We concluded there are no significant discrepancies in the morphology of investigated materials. The only difference, which is however disputable, is the increased tendency towards agglomeration in the case of the ternary hybrid (cf. Supplementary Fig. 17c or 17d with 17a or 17b). We complemented the SEM images with the energy dispersive X-ray analysis (EDS), which confirmed the composition of samples.

In order to elucidate the LIIPS mechanism the light intensity-dependant electrochemical impedance spectroscopy[33] was employed to determine electron lifetimes. The values derived for ZnO fall within the range of 0.5–1.17 s (which corresponds well with previously reported data[34]) and the impact of photoaging is virtually negligible. Consistently with the previous studies, the shorter lifetimes are observed with the increasing incident light intensity[35,36].

The surface modification of ZnO with CA molecules results in a slight decrease of the value for the lowest photon flux, which is in line with expectations—an introduction of the modifier exhibiting a strong acceptor character should facilitate the electron withdrawal with the possibility for some secondary processes. When the light intensity increases the lifetimes follow to reach the value comparable to the neat ZnO—most likely due to the limited number of molecules coordinated to the ZnO surface and the saturation of processes CA is involved in.

Interestingly, for the pre-treated binary hybrid the electron lifetime at the lowest photon flux is over two-times longer than for the fresh sample. It actually shines a light on how the first stage of CA–COH covalent bond formation may look like (cf. "Discussion" section)—a possible explanation involves a partial reduction of CA with electrons from the ZnO conduction band,

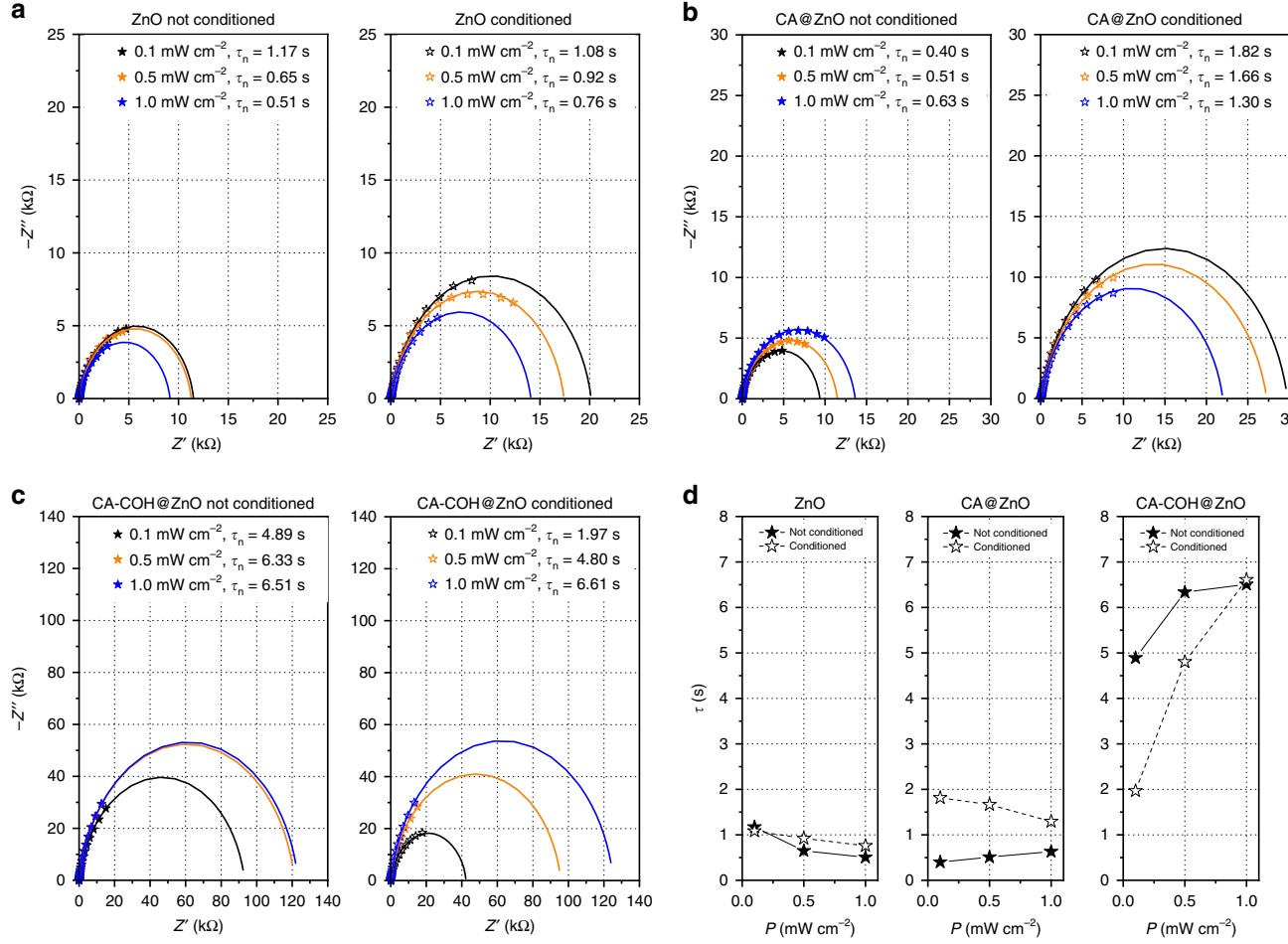

**Fig. 3 The light intensity-dependent electrochemical impedance spectra.** The spectra recorded for ZnO (**a**), CA@ZnO (**b**) and CA-COH@ZnO (**c**) samples before (left) and after the conditioning step (right) are complemented with the simulated curves. The materials were irradiated at three light intensity levels with the applied potential equal to −300 mV vs. Ag/AgCl in air-equilibrated 0.1 mol dm$^{-3}$ KNO$_3$ solution. The determined charge carrier lifetimes (**d**) depend significantly on the incident light intensity in the case of ternary hybrid subjected to the conditioning step.

which shifts its character towards more electron-donating one, hampers secondary processes CA is responsible for and blocks, to some extent, the interfacial electron transfer.

The most striking change occurs, when COH is present in the system. For the fresh sample the lifetime is over fourfold longer than in the case of either neat ZnO or CA@ZnO hybrid and increases slightly with the photon flux—this is probably related to electrons become trapped within COH. Surprisingly though, no LIIPS effect has been observed for the untreated system, which may suggest COH linked through hydrogen bonds with CA is inefficient in the process of oxygen reduction.

After the conditioning step, the value at lower intensities falls to the level observed for the CA@ZnO system, but becomes much longer (up to 6.6 s) at the light intensity of 1 mW cm$^{-2}$. We assume that a suitable pathway for the interfacial electron transfer becomes available when COH binds covalently with CA. As a result, COH serves effectively as a buffer for electrons at lower light intensities and a "fast" oxygen reduction is favoured, yielding cathodic photocurrents. At the same time, the higher photon flux leads to the COH electronic states filling, which in turn allows electrons to be transferred towards the substrate, resulting in the anodic response.

The presented discussion provides a clear evidence for a complex, light intensity-dependent behaviour of the ternary hybrid. Along with the photocurrent polarity change the results indicate a competitive character of two interfacial electron transfer processes characterized by significantly different rate constants, suggesting two distinctive chemical reactions taking place at the surface. The most likely ones involve oxidation of water and reduction of oxygen (vide infra).

## Discussion

From the analysis of samples behaviour in the absence of molecular oxygen we know that cathodic photocurrents, thus also the polarity switching, require oxygen-rich conditions and it occurs only for the ternary system after the pre-treatment procedure (Supplementary Fig. 13f). We have postulated that the addition of both modifiers decreases significantly the charge carriers (electrons) concentration —the conclusion based on slope values determined from the Mott–Schottky plot (Supplementary Fig. 8) —but also results in substantially longer lifetimes (Fig. 3). The former observation is consistent with the data from UPS measurements which indicate a negative shift in the Fermi level energy by 0.4 eV (from 1.8 to 1.4 eV) with respect to the HOMO level for the ternary nanocomposite (Supplementary Fig. 9c)—the work function values obtained with this technique are equal to 3.9, 3.0 and 3.1 eV for neat ZnO, CA@ZnO and CA–COH@ZnO samples, respectively. This result is fully consistent with observed photoelectrochemical properties of the investigated materials— the modified ZnO easily drives the reduction of molecular oxygen dissolved in electrolyte when cathodic potentials are applied

(Supplementary Fig. 7d and f), whereas for the neat semi-conductor only anodic photocurrents are recorded in the whole potentials range.

It is a non-trivial task to formulate a self-consistent mechanistic explanation of the observed behaviour, especially in the case of ternary system. We assume that the interplay between kinetic and thermodynamic aspects of the charge carriers transport, in which both modifiers favour electron withdrawal and trapping events and the semiconductor plays a role of the charge carriers generator, is crucial here. First of all, it seems reasonable to conclude, based on the virtual lack of interactions between COH and ZnO, that CA is a linker—it is capable of binding to two other constituents of the hybrid through hydrogen or covalent bond—from the COH side—or a coordination bond—from the ZnO side.

This assumption is consistent with the aforementioned tendency of COH towards agglomeration in the presence of the quinone (Supplementary Fig. 5), the distinctive changes in IR (Supplementary Figs. 3 and 10) and UV–Vis spectra (Supplementary Figs. 6 and 11) recorded for mixtures of modifiers and hybrids with ZnO, as well as numerous reports on quinone derivatives binding to the semiconductor surface found in the literature. Moreover, the UV–Vis spectra correspond very well with the DFT calculations (Supplementary Fig. 4), which additionally indicate the most feasible transformation that occurs when the sample is irradiated—we assume that the CA molecule anchored to the ZnO surface becomes chemically active, as the result of a partial reduction with electrons from the ZnO conduction band. At the same time, COH may be striped of hydroxyl moieties when subjected to near-UV irradiation[32], which in turn leads to an increase in its electron accepting character[37,38]. As the consequence, the formation of a new covalent bond between CA and COH takes place.

Taking into account relatively strong electron acceptor character of the modifiers and the changes in work function values we postulated that in the first stage of electron transport it is withdrawn from the conduction band of ZnO to the energy level originating from the CA–COH conjugate. In the next step, it becomes trapped in the reservoir of electronic states which may be attributed to COH— hence the significant increase in the electron lifetimes for the pre-treated ternary hybrid (Fig. 3). At

this point, trapped electrons may recombine with holes in the valence band, can be transferred into the substrate or towards the electrolyte depending on the applied electrode potential and the resulting driving force. We may notice from the evolution of photocurrent maps (Supplementary Fig. 7b, d and f) —which are literally constructed from snapshots of the samples response in a state of the thermodynamic equilibrium—that upon modification with CA and CA–COH system the efficiency of the cathodic photocurrent generation process rises at the negative electrode potentials. It has been previously reported[13] that the presence of quinone derivatives may facilitate electron transfer to oxygen species in electrolyte, thus the photocurrent map profile from Supplementary Fig. 7d is easily explainable.

Another interesting thing—which may be easily overlooked—happens when the fullerene derivative is present in the system. As we have already mentioned, a slight anodic shift of the switching potential in favour of cathodic polarization can be noticed—an indication that the electron transfer towards the electrolyte is promoted for a broader range of electrode potentials. More subtle, but still very important variation happens in the anodic region, where the profile of the slope is perturbed. We assume it is caused directly by a competition between typically observed anodic current and enhanced—in the case of the ternary hybrid—cathodic response, a situation similar to the ones discussed in our previous works[13,39].

Here, the kinetic part kicks in—when the light flux is low, the majority of trapping states remain vacant and can participate in the electron transfer processes tipping the scales in favour of the cathodic photocurrent generation involving oxygen species in the electrolyte (Fig. 4a—pathway $3 \rightarrow 3'/3'' \rightarrow 3'''$). The majority of charge carriers are withdrawn from the conduction band to effectively empty (due to the low concentration of excited electrons resulting from the low photon flux) trapping states associated with COH and subsequently to dissolved oxygen as a final electron acceptor, leading to a purely cathodic response. When the light intensity increases, these processes become saturated due to the limited number of available states/molecules coordinated to the ZnO surface. At the sufficiently high photon flux all trapping states are occupied and the anodic photocurrent dominates (Fig. 4b), following the pathway $2 \rightarrow 2'$, accompanied with water oxidation. In both cases, i.e. oxygen reduction and surface

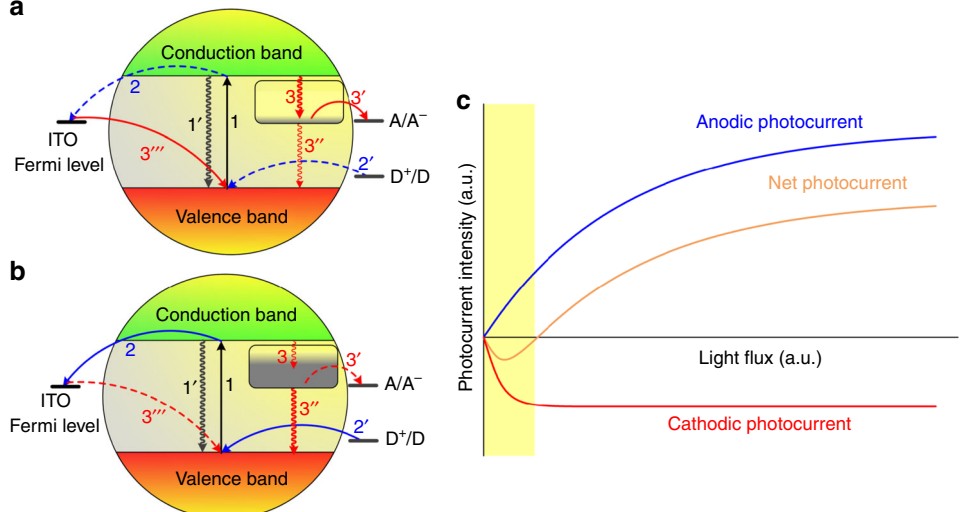

**Fig. 4 The mechanism responsible for Light Intensity Induced Photocurrent Switching (LIIPS).** The photocurrent generation at a semiconducting electrode in the case of low (**a**) and high (**b**) photon flux. The modelled dependence of photocurrent constituents and net photocurrent intensity as a function of the light flux (**c**). The dominating processes are marked as solid arrows and the secondary ones with dashed arrows. The area of cathodic polarization dominance in shaded in yellow.

hydroxyl group oxidation, one-electron processes should be considered as the most feasible ones as follows[40,41]

$$e^-_{CB} + O_2 \rightarrow O_2^{\bullet -} \qquad (1)$$

$$h^+_{VB} + OH^- \rightarrow OH^\bullet \qquad (2)$$

The standard electrochemical potentials of the processes (1) and (2) are equal to $-0.33$ and $2.03$ V (vs. NHE at pH $= 6$), respectively[42]. Taking into account the conduction band potential of the modified ZnO, which is equal to $-0.41$ V (vs. NHE; Supplementary Fig. 8) and the valence band potential of $2.79$ V (vs. NHE), both reactions are thermodynamically favoured.

Additionally, based on the electron lifetimes comparison (Fig. 3) we may conclude, that in the light-soaked ternary system a rather slow recombination occurs and the discharging of the sample is heavily dependent on the interfacial electron transfer processes. It is most likely a consequence of the additional electron traps presence, which may be attributed to the introduction of modifiers. It can be assumed that the photocurrent intensity for a single photoelectrochemical pathway involving a limited number of states can be expressed as (3)[43] due to saturation effects[44–46]:

$$i = i_0\left(1 - e^{-k\phi}\right) \qquad (3)$$

where $i_0$ is the current limit, $\phi$ is the light flux and $k$ is the proportionality factor associated with the kinetic parameters of the process. This approach is consistent with generalized kinetic models of the photocatalytic reaction[47]. A simple numerical experiment based on these assumptions (Fig. 4c) yields results, which correlate nicely with the experimental data.

Our contribution presents a counterintuitive dependence of the photocurrent generation efficiency on the light intensity. Our observations and the model built upon them, when confronted with the studies in the field of photovoltaics and photocatalysis prove, that the significance of light intensity is strongly underestimated in those investigations, which in turn may affect the performance of designed devices. The presence of charge carrier traps—a common feature of many semiconductors, especially those of nanoscale comminution—is responsible for a complex dynamic response to light, which in the extreme case can lead to the reversal of the photocurrent polarity. Solar cells are usually optimized for the unidirectional electron flow, so the presented effect is of less importance, but it seems to be crucial for photocatalytic processes, which involve complex interfacial electron and hole transfer processes.

The involvement of charge-trapping phenomena, which can selectively favour individual electron transfer pathways, is of a profound importance in the photocatalysis, especially in the presence of co-catalysts. Two experimental facts related to the discussed LIIPS effect have been already reported: decreased yields of photocatalytic reactions at the low photon flux[11] and a decrease in the efficiency with an increasing loading of co-catalyst[48,49]. The latter effect, usually described with a volcano curve[50], also results from the competition between various light-induced processes at the surface of a complex composition.

The discussed phenomenon was observed for a relatively complicated system, in which a set of charge-trapping entities was deliberately incorporated, but the mechanism behind it should be common for the majority of nanoscopic semiconducting systems capable of interactions with light. Therefore, a design of new photocatalytic or photovoltaic systems should take into account a complex interplay between various charge trapping, recombination and interfacial processes, as well as associated kinetic effects and thermodynamic parameters also in the context of varying light intensities.

## Methods

**Synthesis of materials.** Fullerenols were synthesized from fullerenes C$_{60}$ (98% Sigma-Aldrich). 50 mg of fullerenes were dissolved in 50 ml of toluene followed by the addition of 2 ml of 2 mol dm$^{-3}$ NaOH, 2 ml of tetra-butyl-ammonium hydroxide (TBAH, 40% in water Sigma-Aldrich) and 5 ml of 30% H$_2$O$_2$. The mixture was stirred for 4 days at ambient temperature. Toluene was removed with the use of a separation funnel and 20 ml of methanol were added yielding yellow precipitate. The material was centrifuged for 10 min at 5000 rpm. Afterwards, the product was washed with methanol and centrifuged again five times. Then, the precipitate was dried at room temperature.

**Synthesis of hybrid materials.** Fifty milligrams of fullerenols (COH) were dissolved in 50 ml of distilled water and sonicated for 20 min. Next, 0.2 mmol dm$^{-3}$ of CA solution in distilled water was added. The mixture was homogenized for 0.5 h. In the next step, 0.3 g of ZnO was added to 20 ml of the solution and it was further diluted with 20 ml of distilled water and the homogenization was repeated for 30 min at the room temperature. The procedure leads to the saturation of accessible adsorption sites for the selected set of conditions, thus it ensures reproducibility. At the same time, due to the complex interactions between constituent of the hybrid, the quantitative determination of the surface coverage is impossible.

**Measurements.** The UV–Vis spectra of modifiers dissolved in water were recorded with the use of an Agilent 8454 UV–Vis spectrophotometer. ATR-FTIR spectra were recorded for powder samples deposited onto a diamond crystal with the use of an FTIR Bruker TENSOR II spectrometer within the range of 350–4000 cm$^{-1}$.

The optical properties of binary and ternary hybrids were determined based on the diffusive reflectance spectrophotometry within the range of 220–2200 nm. The measurements were carried out using Lambda 750 (Perkin Elmer). Each sample was dispersed in spectrally pure BaSO$_4$.

The geometry optimization and molecular orbitals energy were calculated with the use of density functional theory (DFT) methods with the B3LYP hybrid functional and the 3–21G(6D, 7F) basis set. The electronic transitions were determined with the time-dependent (TD-DFT) using the B3LYP functional and the 3–21G(6D, 7F) basis set. First 90 excitations were taken into account. All calculations have been performed using Gaussian09 software package[51]. The GaussView5 programme was used to establish the initial geometry of fullerenols and for the visualisation of the orbitals[52].

Electrodes for the Mott–Schottky analysis were drop-casted onto the surface of ITO@PET. Ag/AgCl (3.5 M KCl) electrode was used as the reference electrode. Pt wire gauze electrode was applied as the counter electrode. The impedance measurements were performed with the use of Biologic SP-200 potentiostat/galvanostat (Bio-Logic Science Instruments, France) equipped with the EIS module. Each electrode was kept at the corresponding potential for 5 s before the measurement. Data points for selected potentials were recorded at the frequency of 1 kHz with the modulation signal amplitude of 5 mV in 0.1 mol dm$^{-3}$ KNO$_3$. The measurements were carried out in the dark.

The DPV curves were recorded using the same setup as in the case of the Mott–Schottky analysis. The measurements were carried out in air-equilibrated 0.1 mol dm$^{-3}$ KNO$_3$ solution at the sweep rate of 10 mV s$^{-1}$.

Electrodes for the photoelectrochemical measurements were drop-casted onto the surface of ITO@PET. Ag/AgCl (3.5 M KCl) electrode was used as the reference electrode. Pt wire was applied as the counter electrode. The photocurrent action maps were recorded using a photoelectric spectrometer (Instytut Fotonowy, Poland) composed of a 150 W xenon lamp, a motorized monochromator with a shutter and a set of cut-off filters and a high precision potentiostat. 0.1 mol dm$^{-3}$ KNO$_3$ solution of pH $= 6$ was used as the electrolyte and it was saturated with either oxygen or argon. All the electrodes were irradiated from the substrate side.

All the LIIPS-related measurements (including the conditioning step) were conducted with the use of Biologic SP-150 potentiostat/galvanostat (Bio-Logic Science Instruments, France) in the three-electrode setup described above with 0.1 mol dm$^{-3}$ KNO$_3$ equilibrated with air as the electrolyte. For the light soaking treatment the LED controller equipped with LED matrix (365 nm, Instytut Fotonowy, Poland) was used at the output power of 125 mW cm$^{-2}$. In the rest of experiments, a single LED (365 nm, Luxeon Star, Canada) was used as the light source. It was powered through the WA-301 wideband amplifier (Aim-TTi, England). Pulse sequences were generated with DG4062 arbitrary function generator (RIGOL Technologies Inc., USA). All the electrodes were irradiated from the substrate side.

The thickness of drop-casted electrodes was determined using DektakXT (Bruker, USA) stylus profiler working in the N-Lite mode, which ensures very low stylus load (radius 2 µm, load 0.5 mg) and allows non-destructive analysis. The average thickness was equal to 7.3 ± 2.8 µm (mean ± standard deviation). No changes relevant to the occurrence of the LIIPS effect were observed for thickness values varying within the investigated range.

The light intensity of LED diode was measured with Power and Energy Meter PM100USB with S310C sensor (ThorLabs).

The light intensity-dependant electrochemical impedance spectroscopy was performed with the use of Biologic SP-200 potentiostat/galvanostat (Bio-Logic Science Instruments, France) equipped with the EIS module. The measurements were carried out in the photoelectrochemical cell PECC-1 (ZAHNER- elektrik,

Germany) using the standard three-electrode setup (Pt wire as the counter electrode, Ag/AgCl as the reference electrode). A calibrated light source (365 nm diode) controlled by the dedicated Zahner PP211 current source was employed. Each electrode was kept at the corresponding potential for 5 s before the measurement. Spectra for selected potentials were recorded in air-equilibrated 0.1 mol dm$^{-3}$ KNO$_3$ solution for frequencies ranging from 1 kHz to 100 mHz (with 10 points per decade) with the potential amplitude of 5 mV at the constant electrode potential of $-300$ mV vs. Ag/AgCl. All the electrodes were irradiated from the substrate side.

The charge carrier lifetime ($\tau$) was determined based on the model designed as a parallel combination of a CPE (a constant phase element) and a resistor ($R_{rec}$), where the CPE may be associated with the chemical capacitance—$C_\mu$—while $R_{rec}$ corresponds to the recombination resistance. The meaning of $C_\mu$ and $R_{rec}$ may be expressed with Eqs. (5) and (6), respectively. The CPE was chosen instead of a standard capacitance in order to make the model more suitable for porous materials. The electron lifetime may be derived based on the following relation[33]:

$$\tau = R_{rec} \cdot C_\mu \tag{4}$$

$$C_\mu = \frac{e^2 \cdot A \cdot L}{k_B \cdot T} \cdot n_c(E) \tag{5}$$

$$R_{rec} = \frac{k_B \cdot T}{e^2 \cdot A \cdot L} \cdot \tau \cdot n_c^{-1}(E) \tag{6}$$

where $k_B$ is Boltzmann's constant, $e$ is the elemental charge, $T$ is the temperature, $L$ is the film thickness, $A$ is the electrode area and $n_c(E)$ is the conduction band electron density as the function of bias potential.

The photovoltage spectra were recorded on a Kelvin probe-based surface photovoltage spectrometer (Besocke Delta Phi, Germany and Instytut Fotonowy, Poland). The reference electrode was a 2 mm gold grid oscillating at ca. 180 Hz. The thin films of ZnO, CA@ZnO and CA-COH@ZnO were illuminated through the electrode using a 150 W xenon arc lamp and a computer-controlled monochromator. The measurement was performed with the 10 nm step in the range of 300–530 nm.

The scanning electron micrographs were taken on Versa 3d (FEI, Switzerland) scanning electron microscope operating at 10 kV with an Everhart–Thornley detector. The chemical composition of the investigated samples was confirmed using the energy-dispersive X-ray spectroscopy.

The ultraviolet photoelectron spectroscopy (UPS) measurements were performed in a PHI VersaProbeII apparatus (ULVAC-PHI, Chigasaki, Japan) using He I line (21.22 eV) from a UHV gas discharge lamp. The acceleration potential of $-5$ V was applied to the sample leading to a much more pronounced secondary electron cut off (SE cut-off). The work function (measured as the difference between photon energy and SE cut-off position) and the hole injection barrier (given by the difference of the substrate Fermi level and the HOMO onset of the material) were measured by UPS. For each UPS spectrum the emission features originating from the secondary line excitations of the He–I gas discharge were subtracted. Since the actual relative intensities of the satellite excitations depend on the He discharge pressure, the secondary line subspectra were adjusted marginally in terms of intensity to the measured UPS spectrum and subtracted incrementally starting with the highest photon energy satellite. The measurement times were kept as short as possible to avoid any possible degradation of the examined materials during the exposure to high-energy radiation (21.22 eV, i.e. 58.4 nm).

## Data availability

The data that support the findings of this study are available from the corresponding author upon reasonable request.

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

## Acknowledgements

Authors would like to thank Mariusz Hajdyła for his assistance during measurements, Dr. Joanna Kuncewicz and Prof. Kapela Pilaka for fruitful discussions and many valuable advices. Financial support from National Science Centre (Grant no. UMO-2016/21/N/ST3/00469, UMO-2013/11/D/ST5/03010 and UMO-2015/17/B/ST8/01783) is gratefully acknowledged. K.P. would like to thank the Foundation for Polish Science (FNP) for the financial support within START programme. DFT calculations were performed at the Academic Computer Centre CYFRONET AGH within computational grant GRAPHENE5.

## Author contributions

A.P. and K.P. developed the concept and performed the first experiments. M.S. was responsible for the chemical preparation of the samples. A.P. carried out most of the photoelectrochemical and spectroscopic measurements and was involved in the DFT simulations. K.M. conducted the Mott–Schottky analysis, the light intensity-dependent EIS experiments and DPV measurements. M.M. recorded and interpreted UPS data. K.P. recorded SEM images and EDS spectra. K.S. took part in the mechanism formulation, the DFT modelling and results interpretation. K.P. coordinated the experimental work, interpreted the results and wrote the manuscript in consultation with the other authors.

## Competing interests

The authors declare no competing interests.
