## [Peer Review File · Nature Communications]

Reviewers' comments:

Reviewer #1 (Remarks to the Author):

This is an interesting paper that claims new photocurrent switching effects induced by light intensity modulation instead of potential change. The manuscript brings novel information and could reach a large community. There are strong positive aspects (clear experimental evidence of photocurrent switching; convincing combination of experimental characterization techniques). However, my opinion is that the paper cannot be accepted as it is; several aspects need further clarification before a final decision is taken:

Major issues

1. Page 4: The reversibility of the phenomenon reported on Figure 1 (and on Figure 3) is not clearly discussed. It should be specified whether the light power density was decreased or increased during the experiment and if the same effect would be observed when power density is reversed.
2. It is written (page 20, line 395) that during the UPS tests, the measurement times were kept as short as possible to avoid UV damage, and at the same time, the near-UV treatment is a prerequisite to observe photocurrent switching (Page 11, line 204). Is there not a contradiction here? A shortcoming of the manuscript is that there is no indication that the chemistry is maintained after the irradiation. Is there any technique (e.g. an electrochemical signature) that could be used to assess that irradiation does not affect the chemistry of the system and that AC & COH remain chemically unaffected?
3. Also, it is unclear for the reader whether this irradiation has a permanent effect on the LIIP effect or only a temporary effect (may be there is a charge accumulation, proportional to the time of irradiation, that fades away during experiments?).
4. What is the impact of the ZnO/CA/COH proportions on the results? There might be a correlation between a concentration of trapping sites and the duration of the irradiation. A comment would be welcome here.
5. Some information should be provided regarding the microstructure of the hybrid materials. Also, the impact that the irradiation (prior to measurements) can have on the microstructure, if any, should be discussed or at least evoked.
6. Page 8 line 162-164: some very specific switching conditions are identified (365 nm; -300 mV); factors that could change these conditions (hybrid composition, irradiation conditions like the potential of irradiation or the duration) are not clearly identified or discussed. It is not necessary to report a detailed analysis of these factors but to tell to the reader if an impact is expected, in order to show that the observations reported in the manuscript are robust and not an artefact.
7. Page 12, line 237: Is the idea developed here that there is a coexistence of anodic and cathodic photocurrent at the potential of the experiment and that one becomes predominant over the other as a function of light intensity?
8. What is the link between the sign of the photocurrents and the redox processes in solution? Is it water oxidation/reduction? The H₂O/O₂ redox couple is highly irreversible at ambient temperature so it is difficult to admit that anodic and cathodic photocurrents of Figure 1 could be associated to OER and ORR. The H₂O/H₂ redox couple is much more reversible. However, there is a potential mismatch: at pH=7 (KNO₃), $E^{\circ} = -0.42 \text{ V/ESH} = -0.64 \text{ V/AgAgCl}$. Since the experiment reported in Figure 1 was made at a constant potential of -300 mV/Ag-AgCl, it is unlikely that the cathodic photocurrent could be due to the HER. May be I did not get the point but there is a need for clarification from the authors.
9. Page 14, lines 255-257: the conclusion is intended to give a general character to the observations reported in the manuscript but this is not fully convincing because the CA/COH combination was made intentionally to put into evidence the LIIPS effect. What would be the added value of these findings for photocatalysis where co-catalysts are usually deposited at the surface of the semiconductor to fasten the charge transfer kinetics? Is it claimed that such co-catalysts could also induce similar current switching effects?

Minor issues

1. Page 1, line 22 : we also speculate instead of postulate?
2. In the caption of Figure 1, the potential at which the experiment was made should be specified explicitly.
3. Page 4, Figure 1: redox processes occurring during anodic and cathodic photocurrent in such KNO₃ aqueous should be specified explicitly (OER and HER ?).
4. Irradiation conditions prior to measurements are mentioned (e.g. p.8 line 165) but not clearly detailed and justified (e.g. was the potential of 300 mV chosen arbitrarily or is it an optimum value?). They certainly impact significantly the results (as indicated p.4, line 77) and should be described in more details in the experimental section.
5. Page 10 line 177-178: this is therefore a third condition that should be added to the list of prerequisite conditions mentioned page 4 line 76.
6. Page 10 lines 194-196: in terms of perspectives, would it be useful to change the chemistry of the quinone (for ex. to change the potential at which the current switch occurs) ?

Reviewer #2 (Remarks to the Author):

From my point of view, this work does not contain any new physical insight. The paper style seems more like a research report rather than a scientific paper. This work does not contain any new physical insight. I do not find anything new, especially on synthetic procedure or new properties. This paper is not suitable for Nature communications.

Authors have developed a ternary hybrid material composed of ZnO, CA and fullerenols to understand light intensity induced photocurrent switching effect.

Major issues are:

How the defect band of ZnO effects on photocurrent generation? How the thickness plays the role. How CA is attached with ZnO and finally with fullerenols? No experimental evidence is given by authors.

Photocurrent generation is due to charge transfer process. Without giving proper evidence of charge transfer by using spectroscopy, it is not convincing.

Scientific evidence is missing. This work is definitely not suitable for this high impact journal.

The mechanism part is missing.

Electron or hole transfer processes should be explained from decay time measurement.

For this reason, this work is not novel and not significant for publication for this journal.

Reviewer #3 (Remarks to the Author):

The authors A.Podborska, K. Pilarczyk et.al. present a new phenomenon: light intensity induced photocurrent switching (LIIPS), occurring under modulated light intensity on the ternary CA-COH@ ZnO system for switching the photocurrent polarity. The results presented are intriguing and could lead to a high-ranking publication such as the Nature Communications, however, currently there are several inconsistencies which need to be addressed prior any publication.

Although the discovery is undoubtedly new and might be of interest in the many fields, the manuscript contains more questions mark than the statements. At the present stage the experimental data is convincing that indeed the phenomenon occurs under specific conditions, but the mechanism is not sufficiently supported. The following issues are not fully explained and should be readdressed:

1. Presence of oxygen: it is not clear what is the role of oxygen in the solution in regard of LIIPS effect –How does the presence of oxygen affect the cathodic current within the time? Does the reduction of oxygen occur under applied experimental conditions? How would affect the ZnO surface traps saturation (ex. at elevated temperature and in flowing oxygen on the occurrence of the cathodic current? More studies are required here to explain these observations as this has a significant impact on the overall study.

2. Work function vs photopotential: the energy diagram presented in Fig. S6 is not consistent with the Fig. S9. Taking into account that the work function follows the order: 3.9, 3.1, 3.0 for ZnO, CA-COH@ZnO, CA@ZnO, the change of CPD, should follow the same order in the dark. Fig. S9 represents the changes in the photovoltage under specific wavelength but the starting point has been recorded under the dark, and therefore the relative order at the Y axis is not consistent with the value of the work function presented in Fig.S6: the red line should be the “highest” one. This discrepancy makes the claimed mechanism is not fully plausible

Moreover, the placement of the Evac level should be the same for all systems, which is not the case of diagrams presented in Fig. S6. Minor: the subtitle contains two b) instead of b) and c)

3. Ability to reproduce the results: although the mechanism of the observed phenomenon needs to be more justified, the experimental part is the weakest part of the manuscript. It is not likely to reproduce the films based on the provided description. Lack of any SEM or TEM images or any structural investigation questions the morphology and quality of the film.

4. Time-resolved spectroscopic measurements under low intensity irradiation should provide more information about the intermediates and charge carriers dynamics.

5. Minor remark: it is confusing to use a term binary in connection to hybrid composite and modifiers as well.

6. Caption of Fig.1 does not contain any information about the imposed potential (I vs t)

7. The authors give impression to misunderstand the significance of Mott Schottky relation that in the case of a n-type semiconductor allows one to evaluate the density of majority charge carriers (i.e., electrons) but certainly not holes, thus the claim that ‘...the binary modifier decreases significantly the charge carriers concentration’ (line 180) is not precise. Besides, note that the impedance measurements have been performed under dark conditions (unless it was not mentioned), therefore how this observation does affect the authors conclusions regarding the nature of the junction of the binary modifier with ZnO?

In summary, the appearance of the light intensity induced photocurrent switching is unquestionable, however an explanation of the origin of this phenomenon requires more exhausting experimental support and justification. Regrettably I do not recommend this paper for publication in the present version, but it might be suitable for the journal after major revisions.

Point-by-point answers to reviewers' comments:

Reviewer #1:

This is an interesting paper that claims new photocurrent switching effects induced by light intensity modulation instead of potential change. The manuscript brings novel information and could reach a large community. There are strong positive aspects (clear experimental evidence of photocurrent switching; convincing combination of experimental characterization techniques). However, my opinion is that the paper cannot be accepted as it is; several aspects need further clarification before a final decision is taken:

Major issues

1. Page 4: The reversibility of the phenomenon reported on Figure 1 (and on Figure 3) is not clearly discussed. It should be specified whether the light power density was decreased or increased during the experiment and if the same effect would be observed when power density is reversed.

The photocurrent switching effect discussed in the paper is persistent and independent on the light intensity changes direction - an appropriate comment has been added to the text (lines 204 and 209) along with new results (Fig. S14). We also describe other experiments in a bit more detailed manner (cf. the caption of Fig. 1).

2. It is written (page 20, line 395) that during the UPS tests, the measurement times were kept as short as possible to avoid UV damage, and at the same time, the near-UV treatment is a prerequisite to observe photocurrent switching (Page 11, line 204). Is there not a contradiction here ? A shortcoming of the manuscript is that there is no indication that the chemistry is maintained after the irradiation. Is there any technique (e.g. an electrochemical signature) that could be used to assess that irradiation does not affect the chemistry of the system and that AC & COH remain chemically unaffected?

The conditioning step and LIIPS experiments are carried out with the use of UV radiation characterized by relatively low energy (3.40 eV, i.e. 365 nm), whereas the UPS measurements are performed at much higher energies (21.22 eV, i.e. 58.4 nm). Nonetheless, the lapidary description we used (old line 395) may be confusing. Now we put emphasis on the high energy of radiation used in the UPS measurements (line 608). Based on the new experimental results and DFT calculations we now discuss probable changes in the chemistry of CA and COH in more details (e.g. line 118 and 289 and Fig. S4 and S6).

3. Also, it is unclear for the reader whether this irradiation has a permanent effect on the LIIP effect or only a temporary effect (maybe there is a charge accumulation, proportional to the time of irradiation, that fades away during experiments?).

The charge accumulation process should be excluded – the conditioning step alters irreversibly how the hybrid responses to light. We even tried to reverse these changes by subjecting the pre-treated sample to cyclic voltammetry measurements (Fig. S15), which resulted in some variations in the output for the intermediate light intensities (for which the competition between postulated processes can be easily perturbed), but the LIIPS effect was

still observed. We have also added an appropriate comment on this subject to the text (lines 200-216).

4. What is the impact of the ZnO/CA/COH proportions on the results? There might be a correlation between a concentration of trapping sites and the duration of the irradiation. A comment would be welcome here.

In order to obtain the ternary hybrid we have allowed modifiers to reach an equilibrium with the ZnO surface for the chosen conditions. This approach ensures reproducibility. At the same time, due to the complex nature of interactions between constituents of the composite, it is a difficult task to quantitatively determine the surface coverage – we have added an appropriate remark in the Methods section (line 523). We also commented on the irradiation time within the pre-treatment step (line 200). We believe ZnO/CA/COH ratio plays a role in the hybrid material behaviour, but a detailed, systematic study on this matter would be extremely time-demanding and outside the scope of this contribution (the aim of which is to present a new, interesting effect).

5. Some information should be provided regarding the microstructure of the hybrid materials. Also, the impact that the irradiation (prior to measurements) can have on the microstructure, if any, should be discussed or at least evoked.

We fully agree. An appropriate paragraph has been added to the text (line 217) along with the SEM images and the EDS analysis (Fig. S17).

6. Page 8 line 162-164: some very specific switching conditions are identified (365 nm; -300 mV); factors that could change these conditions (hybrid composition, irradiation conditions like the potential of irradiation or the duration) are not clearly identified or discussed. It is not necessary to report a detailed analysis of these factors but to tell to the reader if an impact is expected, in order to show that the observations reported in the manuscript are robust and not an artefact.

At first glance, the described effect may look as an artefact (because it is somehow counterintuitive). In the course of experimental work, we managed to get repeatable results for a certain set of parameters and these are presented as the initial conditions. Nonetheless, we later verified to what extent some of them could be varied. In the revised version of the manuscript we present a map (Fig. 2), which depicts the electrode potentials that may be used in both the conditioning step and during the LIIPS measurements (line 192). We also comment on other aspects, such as the role oxygen plays in the photocurrent switching (e.g. line 266), or the irradiation time (line 200). Moreover, the information on the incident light wavelengths range that can be used, although not provided outright, may be acquired from the photocurrent action maps (Fig. S7) and the relevant discussion in the text.

7. Page 12, line 237: Is the idea developed here that there is a coexistence of anodic and cathodic photocurrent at the potential of the experiment and that one becomes predominant over the other as a function of light intensity?

Yes, the LIIPS mechanism is attributed to the competitive generation of anodic and cathodic photocurrents – an effect we have observed also in the case of other hybrid materials. A more detailed discussion, along with possible interfacial redox reactions, is added to the text (line 331).

8. What is the link between the sign of the photocurrents and the redox processes in solution? Is it water oxidation/reduction? The H₂O/O₂ redox couple is highly irreversible at ambient temperature so it is difficult to admit that anodic and cathodic photocurrents of Figure 1 could be associated to OER and ORR. The H₂O/H₂ redox couple is much more reversible. However, there is a potential mismatch: at pH=7 (KNO₃), E° = - 0.42 V/ESH = -0.64V/AgAgCl. Since the experiment reported in Figure 1 was made at a constant potential of -300 mV/Ag-AgCl, it is unlikely that the cathodic photocurrent could be due to the HER. Maybe I did not get the point, but there is a need for clarification from the authors.

Although the electrode is maintained at -0.3 V vs. Ag/AgCl, the potentials of electrons in the conduction band and holes in the valence band are equal to -0.41 V and +2.79 V vs. NHE, respectively (as derived from the Mott-Schottky analysis and the bandgap width). Taking into account the one-electron redox processes at oxide surfaces – oxygen reduction at -0.33 V and water oxidation at 2.03 V vs. NHE – both reactions are thermodynamically favoured. The initial submission lacked the detailed analysis on this matter – we have added an appropriate section explaining the system thermodynamics (line 331).

9. Page 14, lines 255-257: the conclusion is intended to give a general character to the observations reported in the manuscript but this is not fully convincing because the CA/COH combination was made intentionally to put into evidence the LIIPS effect. What would be the added value of these findings for photocatalysis where co-catalysts are usually deposited at the surface of the semiconductor to fasten the charge transfer kinetics? Is it claimed that such co-catalysts could also induce similar current switching effects?

We have some premises to assume the described phenomenon occurs also in other systems (including neat semiconductors), in which the alignment of trapping states is appropriate; thus we allowed ourselves to speculate that the LIIPS effect is of a more general character and the CA-COH@ZnO hybrid is not an excluded example. To support the statement we have included a short discussion in the Introduction (line 48) and in the last paragraphs of the manuscript.

Minor issues

1. Page 1, line 22 : we also speculate instead of postulate?

We softened this statement.

2. In the caption of Figure 1, the potential at which the experiment was made should be specified explicitly.

We added the appropriate information.

3. Page 4, Figure 1: redox processes occurring during anodic and cathodic photocurrent in such KNO₃ aqueous should be specified explicitly (OER and HER ?).

We now specify them in the Discussion section.

4. Irradiation conditions prior to measurements are mentioned (e.g. p.8 line 165) but not clearly detailed and justified (e.g. was the potential of 300 mV chosen arbitrarily or is it an optimum value?). They certainly impact significantly the results (as indicated p.4, line 77) and should be described in more details in the experimental section.

We attached a map (Fig. 2) indicating the potential ranges (for both the conditioning step and further measurements), within which the LIIPS effect occurs for the investigated system. The revised Methods section provides a full description of the LIIPS-related measurements (line 558).

5. Page 10 line 177-178: this is therefore a third condition that should be added to the list of prerequisite conditions mentioned page 4 line 76.

We corrected that part.

6. Page 10 lines 194-196: in terms of perspectives, would it be useful to change the chemistry of the quinone (for ex. to change the potential at which the current switch occurs)?

It would be definitely interesting, although it is hard to predict how the interaction with fullerenols would be affected. Nevertheless, we plan to investigate other hybrid materials containing various carbon nanostructures in terms of the LIIPS effect appearance. At this stage, however, the discussion on the quinone chemistry modifications is outside the scope of this manuscript.

Reviewer #2:

From my point of view, this work does not contain any new physical insight. The paper style seems more like a research report rather than a scientific paper. This work does not contain any new physical insight. I do not find anything new, especially on synthetic procedure or new properties. This paper is not suitable for Nature communications.

The aim of this study was not to develop a new synthetic procedure, but to present a new, interesting effect – we have strong premises to believe, the phenomenon we are discussing is new and has not been reported elsewhere.

Authors have developed a ternary hybrid material composed of ZnO, CA and fullerenols to understand light intensity induced photocurrent switching effect. Major issues are: How the defect band of ZnO effects on photocurrent generation?

As it is discussed in the first paragraph of “The hybrid materials” subsection, ZnO does not exhibit any cathodic response to the irradiation. In contrast to other (heavily defected) forms of zinc oxide (e.g. RSC Adv. 2014, 4, 58553), the investigated material does not exhibit any photoelectrochemical activity within sub-bandgap excitations (cf. Fig. S7a, S7b and S11). It may be therefore concluded, there is no defect band involved in the photocurrent generation process. Furthermore, even if there are some photoelectrochemically silent defect states, they should affect the response of both modified and unmodified material. In the studied case, neither neat ZnO nor the semiconductor modified with chloranilic acid exhibit any signs of the LIIPS effect; hence any measurable influence of the defect band can be excluded.

How the thickness plays the role.

In all the experiments involving the interaction with light, electrodes are irradiated from the substrate side in order to reduce a possible influence the thickness has on the result. An appropriate comment has been added to the Methods section.

How CA is attached with ZnO and finally with fullerenols? No experimental evidence is given by authors.

We comment on this matter extensively in several places. We have also managed to tie up loose ends with the use of DFT modelling, IR and UV-Vis spectroscopy – please refer to “The interplay between modifiers” section (line 102) – and the paragraph in line 286.

Photocurrent generation is due to charge transfer process. Without giving proper evidence of charge transfer by using spectroscopy, it is not convincing. Scientific evidence is missing. This work is definitely not suitable for this high impact journal.

The electron transfer process should be understood here in a more general manner and should not be confused with the charge-transfer phenomenon comprehended in a spectroscopic sense. As we do not observe any significant photosensitization (line 162), we can assume the photocurrent generation is due to the fundamental transition (i.e. the electron excitation from the valence to the conduction band of the semiconductor occurs). Fig. S11 clearly shows the absorption onset below 400 nm, which in the case of zinc oxide should be associated with the fundamental transition.

The mechanism part is missing. Electron or hole transfer processes should be explained from decay time measurement. For this reason, this work is not novel and not significant for publication for this journal.

We have decided to use a method based on the electrochemical impedance spectroscopy in order to determine electron lifetimes – please refer to Fig. 3 and an associated discussion (line 230 onwards).

Reviewer #3:

The authors A.Podborska, K. Pilarczyk et al. present a new phenomenon: light intensity induced photocurrent switching (LIIPS), occurring under modulated light intensity on the ternary CA-COH@ZnO system for switching the photocurrent polarity. The results presented are intriguing and could lead to a high-ranking publication such as the Nature Communications, however, currently there are several inconsistencies which need to be addressed prior any publication.

Although the discovery is undoubtedly new and might be of interest in the many fields, the manuscript contains more questions mark than the statements. At the present stage the experimental data is convincing that indeed the phenomenon occurs under specific conditions, but the mechanism is not sufficiently supported. The following issues are not fully explained and should be readdressed:

1. Presence of oxygen: it is not clear what is the role of oxygen in the solution in regard of LIIPS effect –How does the presence of oxygen affect the cathodic current within the time? Does the reduction of oxygen occur under applied experimental conditions? How would affect the ZnO surface traps saturation (ex. at elevated temperature and in flowing oxygen on the occurrence of the cathodic current? More studies are required here to explain these observations as this has a significant impact on the overall study.

During the photoelectrochemical measurements the photoelectrode is maintained at the potentials falling within the range of $+0.3 \div -0.5$ V vs. Ag/AgCl. At the same time, the potentials of electrons in the conduction band and holes in the valence band are equal to -0.41 V and $+2.79$ V vs. NHE, respectively (as derived from the Mott-Schottky analysis and the bandgap width). Therefore oxygen reduction is thermodynamically favoured during the electrode irradiation (the standard potential of -0.33 V vs. NHE, cf. line 331). The role of the doping was not studied (it is outside the scope of the presented contribution), as we have focused on the influence of surface modifications.

2. Work function vs photopotential: the energy diagram presented in Fig. S6 is not consistent with the Fig. S9. Taking into account that the work function follows the order: 3.9, 3.1, 3.0 for ZnO, CA-COH@ZnO, CA@ZnO, the change of CPD, should follow the same order in the dark. Fig. S9 represents the changes in the photovoltage under specific wavelength but the starting point has been recorded under the dark, and therefore the relative order at the Y axis is not consistent with the value of the work function presented in Fig.S6: the red line should be the “highest” one. This discrepancy makes the claimed mechanism is not fully plausible Moreover, the placement of the Evac level should be the same for all systems, which is not the case of diagrams presented in Fig. S6. Minor: the subtitle contains two b) instead of b) and c)

We overlooked this incoherence in a way the data was presented in Fig. S9 (now Fig. S12) – the work function determination utilizing the Kelvin probe (KP) technique demands some sort of reference to be used, which ensures the obtained value may be treated in an absolute manner. In the case presented, we were interested in relative changes between the investigated

samples and no reference material was used; thus the CPD values in the dark (first data points) cannot be directly compared to the absolute values derived from the UPS measurement. We modified Fig. S12 accordingly to avoid any confusion. As to the Evac level – two approaches are typically applied to the construction of energy diagrams – the first one assumes the Evac level is constant and the second one align the Fermi levels. The first scenario is more common in the case the KP technique is employed, whereas the second one is often used in the case of the UPS measurements.

3. Ability to reproduce the results: although the mechanism of the observed phenomenon needs to be more justified, the experimental part is the weakest part of the manuscript. It is not likely to reproduce the films based on the provided description. Lack of any SEM or TEM images or any structural investigation questions the morphology and quality of the film.

All experimental details, as well as structural characterization of the studied materials have been added to the revised version of the manuscript. The SEM images and the EDS analysis are presented in the supplementary materials (Fig. S17) and an appropriate comment is present in the text (line 217).

4. Time-resolved spectroscopic measurements under low intensity irradiation should provide more information about the intermediates and charge carriers dynamics.

The electrochemical impedance spectroscopy performed at various light intensities has provided significant information about the electron lifetimes in the investigated systems – please refer to Fig. 3 and an associated discussion (line 230 onwards). The collected data confirm previously postulated mechanism behind the LIIPS effect.

5. Minor remark: it is confusing to use a term binary in connection to hybrid composite and modifiers as well.

The descriptions of samples and materials have been carefully corrected to avoid any further confusions.

6. Caption of Fig.1 does not contain any information about the imposed potential (I vs t).

The figure caption has been corrected as requested

7. The authors give impression to misunderstand the significance of Mott Schottky relation that in the case of a n-type semiconductor allows one to evaluate the density of majority charge carriers (i.e., electrons) but certainly not holes, thus the claim that ‘...the binary modifier decreases significantly the charge carriers concentration’ (line 180) is not precise. Besides, note that the impedance measurements have been

performed under dark conditions (unless it was not mentioned), therefore how this observation does affect the authors conclusions regarding the nature of the junction of the binary modifier with ZnO?

The statements based on the electrochemical impedance spectroscopy measurements have been corrected (lines 154 and 269). New results, collected during EIS experiments under illumination have been also presented and discussed. The EIS experiments carried out in the dark were used in order to indicate the change in the flat-band potential value and electron concentration induced by the introduction of the modifiers.

In summary, the appearance of the light intensity induced photocurrent switching is unquestionable, however an explanation of the origin of this phenomenon requires more exhausting experimental support and justification. Regrettably I do not recommend this paper for publication in the present version, but it might be suitable for the journal after major revisions.

An exhaustive mechanistic study (in terms of both experimental and theoretical approach) has been added to the manuscript. All obtained results confirm the initially suggested mechanism, but at the same time, provide a substantial amount of additional information on the chemistry and photophysical properties of the investigated systems.

Reviewers' comments:

Reviewer #1 (Remarks to the Author):

The necessary clarifications have been made and the revised paper is now much better than the initial submission.

I think that the observations reported in the manuscript are robust and not an artefact.

Experimental conditions are now adequately described and the work could be reproduced by others.

My opinion is that the paper is now acceptable for publication in Nature Communications.

P. Millet

Reviewer #2 (Remarks to the Author):

Authors have partially given answers to the questions which are not satisfied.

Still the questions are not properly answered,

1. How the defect band of ZnO effects on photocurrent generation? How the thickness plays the role.

2. How CA is attached with ZnO and finally with fullerenols? No experimental evidence is given by authors.

3. Photocurrent generation is due to charge transfer process. Without giving proper evidence of charge transfer by using spectroscopy, it is not convincing. Scientific evidence is missing. This work is definitely not suitable for this high impact journal.

4. The mechanism part is missing.

5. Electron or hole transfer processes should be explained from decay time measurement.

For this reason, this work is not scientifically enriched which can not be supported for publication for this journal.

Reviewer #3 (Remarks to the Author):

The reviewed version of the previously submitted manuscript on Light Intensity Induced Photocurrent Switching Effect by K. Pilarczyk et al. represents a high quality and experimentally well supported research which deserves to be published in the Nature Communications. The reported phenomena is new, firstly reported and might significantly affect more than only photocatalysis field. Although water splitting is an important and urgent topic in view of hydrogen production and search of new alternative to water oxidation processes for speeding up the water reduction kinetics, production of any other solar fuels p.ex. from CO₂ reduction cannot be neglected. It has already been noticed that gold plasmons induced at very small nanoparticles are able to direct the CO₂ reduction to other than CO reduction products under different light intensity. In this regard, the present research is urgently needed to understand the nature of the light intensity induced phenomena and also to judge the usefulness of this knowledge towards practical implications. As a reviewer, although I do not agree with the authors that the transient spectroscopy investigations might provide similar to impedance measurements (in the dark and under illumination) conclusions, I agree that the performed number of experimental approaches which are in the present version very consistent with the postulated mechanism is fair enough for the Communication form purposes. In connection to the above-mentioned remarks I am satisfied with the changes have been made and recommend this manuscript for publication without any further changes.

Reviewer name: Renata Solarska

Point-by-point answers to reviewers' comments:

Reviewer #2:

Authors have partially given answers to the questions which are not satisfied. Still the questions are not properly answered,

I believe a word of explanation may be needed. In our previous response, we might have not emphasized enough the main idea behind the publication, which is to communicate the very existence of a new phenomenon, significant for the photochemical and photophysical studies on surface modified semiconductors, thus important for the research on photocatalysts and photovoltaic devices, rather than to discuss all the aspects of the investigated system in details. The presented ternary hybrid and the mechanistic consideration of the LIIPS effect should be treated here more like a platform for the demonstration of the phenomenon, rather than a goal of the study in itself.

1. How the defect band of ZnO effects on photocurrent generation?

First of all, we have investigated the photocurrent response of the studied systems in a broad range of potentials and irradiating light wavelengths. In the case of neat ZnO we have observed no cathodic photocurrents (cf. Fig. S7a and S7b) and no sub-bandgap absorption (cf. Fig. S11), the observation that suggests no significant contribution of defect states to the photoelectrochemical activity (cf. RSC Adv., 2014, 4, 58553). The existence of a defect band may be investigated based on the spectrofluorometry measurements (cf. Nanoscale Res. Lett., 2007, 2, 297); the emission spectra (Fig. R1) recorded for the studied zinc oxide sample match the PL profiles found in other reports (cf. Nanoscale, 2016, 8, 7631) and provide an evidence for the presence of defect bands and their impact on the luminescence response of the material – a broad emission band at approx. 520 nm.

Figure R1. The emission spectra recorded for aqueous suspensions of the investigated materials. The excitation wavelength was 320 nm.

Noteworthy, the emission is quenched in the presence of modifiers, which may be attributed to their acceptor character, promoting the mechanism depicted in Fig. 4, rather than the radiative recombination process – the observation which is fully consistent with the explanation presented in the article.

How the thickness plays the role.

As it is stated in the Methods section the materials were drop-casted onto ITO@PET substrates – the deposition method which obviously does not provide the best level of control over the thickness. Nevertheless, this parameter does not affect the occurrence of the LIIPS effect. In the applied experimental setup the electrode is irradiated through the PET substrate, so due to the limited light penetration depth (not exceeding 100 nm) at the investigated wavelength (i.e. 365 nm; cf. *Int. J. Electrochem.*, 2011, 563427 and *Phys. Rev. B*, 1998, 58, 3586) and the average crystallite size (cf. Fig. S17) it is unlikely that ZnO layers not directly attached to the ITO@PET surface would contribute significantly to the electrode response.

2. How CA is attached with ZnO and finally with fullerenols? No experimental evidence is given by authors.

We comment on this matter extensively in “The interplay between modifiers” section, where we refer to the FTIR spectra (Fig. S3), in which a significant shift of a band attributed to the C=O bond stretching mode in CA may be noticed in the presence of COH. Then we move to the analysis of UV-Vis spectra recorded in various conditions, based on which we propose a mechanism of interaction between CA and COH (cf. Fig. S5 and S6), that we validate based on the comparison of experimental results with the DFT models (cf. Fig. S4) – these are in a full agreement. The main conclusion drawn from the data is that the modifiers exhibit a tendency towards hydrogen bonds formation, and upon irradiation a covalent bond is formed between COH and CA.

In the second paragraph of “The hybrids materials” section we discuss how the modifiers interact with the ZnO surface. We explicitly deny any significant interactions between zinc oxide and COH alone. We also suggest CA molecules coordinate to the ZnO surface through adjacent hydroxyl and carbonyl groups, the assumption for which we find evidences in the FTIR spectra recorded for the hybrids (cf. Fig. S10) and in the literature – the coordination mode should be somehow similar to interactions between surfaces of oxide semiconductors and quinone/anthraquinone derivatives or catechols (cf. Ref. 12, 13, 29, *Angew. Chem. Int. Ed.* 2014, 53, 6322, *Chem. Mater.* 2015, 27, 358, *RSC Adv.*, 2015, 5, 106877 and *Angew. Chem. Int. Ed.* 10.1002/anie.201801063).

3. Photocurrent generation is due to charge transfer process. Without giving proper evidence of charge transfer by using spectroscopy, it is not convincing. Scientific evidence is missing. This work is definitely not suitable for this high impact journal.

The photocurrent generation results directly from various processes involving charge carriers transport – as a matter of fact, this statement is valid for any electric current flow – and only

a small fraction of possible mechanisms responsible for the photocurrent generation can be connected with the charge-transfer phenomenon occurring within an electron-donor-acceptor complex with resulting CT bands, that may be observed with the use of UV-Vis spectroscopy (cf. *Coord. Chem. Rev.*, 2016, 325, 135, *J. Mater. Chem. A*, 2013, 1, 7816, *J. Phys. Chem. C*, 2008, 112, 9530, *Phys. Rev. Lett.*, 1998, 80, 821, *Nat. Photonics*, 2014, 8, 47 and *Chem. Commun.*, 2015, 51, 12286). Actually, in the majority of cases the fundamental transition (i.e. the electron excitation from the valence to the conduction band of a semiconductor) can be identified as a main source of electron-hole pairs which contribute to the light-driven current flow.

We report in the manuscript that no significant photosensitization could be noticed (cf. Fig. S7, S11 and line 162), a process typically observed in the case of a CT complex involvement in the photocurrent generation. Hence, it is safe to assume the photocurrent generation must be related to the fundamental transition within the semiconductor and subsequent charge carriers transfer processes within the system, which should definitely not be confused with the spectroscopically active electron-donor-acceptor complex formation.

4. The mechanism part is missing.

We agree that the mechanistic part may be overlooked, as it is spread over the whole manuscript and various aspects of the mechanism responsible for the LIIPS effect occurrence are discussed in a contextual manner accompanied by relevant experimental data. However, we also devoted almost whole Discussion section (between lines 279 and 350) to the elucidation of the mechanism. We have also double-checked all the details and the presented reasoning seems to be self-consistent and agrees well with all the obtained results. We also plan to study similar systems, focusing on the LIIPS effect occurrence, in order to create a generalised description of the phenomenon, but this is outside the scope of this contribution.

5. Electron or hole transfer processes should be explained from decay time measurement.

We have decided to use a method based on the electrochemical impedance spectroscopy in order to determine electron lifetimes – please refer to Fig. 3 and an associated discussion (line 230 onwards) – instead of using transient photocurrent techniques (TPC). One of the reasons for this choice was to avoid possible degradation of the ternary material, which could be caused by the irradiation with short light pulses from a high-output source. At the same time, the experimental conditions during the EIS-based measurements are similar to the LIIPS-related tests, which makes finding a correlation between the results easier.

REVIEWERS' COMMENTS:

Reviewer #2 (Remarks to the Author):

From my point of view, this work does not contain any new physical insight. It is a routine work. This work does not contain any new physical insight. I do not find anything new, especially on synthetic procedure or new properties. This paper is not suitable for Nature communications.

This work is definitely not suitable for this high impact journal.

I am not satisfied the answers what they have written.